# Possible Value of Faecal Immunochemical Test (FIT) When Added in Symptomatic Patients Referred for Colonoscopy: A Systematic Review

**DOI:** 10.3390/cancers15072011

**Published:** 2023-03-28

**Authors:** Henrike Jacoba Brands, Brigit Van Dijk, Richard M. Brohet, Henderik L. van Westreenen, Jan Willem B. de Groot, Leon M. G. Moons, Wouter H. de Vos tot Nederveen Cappel

**Affiliations:** 1Gastroenterology and Hepatology, Isala Hospital, 8025 AB Zwolle, The Netherlands; 2Department of Epidemiology and Statistics, Isala Hospital, 8025 AB Zwolle, The Netherlands; 3Abdominal Surgery, Isala Hospital, 8025 AB Zwolle, The Netherlands; 4Isala Oncology Center, Isala Hospital, 8025 AB Zwolle, The Netherlands; 5Gastroenterology and Hepatology, Universitair Medisch Centrum Utrecht, 3584 CX Utrecht, The Netherlands

**Keywords:** colorectal cancer, faecal immunochemical test (FIT), symptoms

## Abstract

**Simple Summary:**

Literature has shown that the correlation between intestinal complaints and the gain of colonoscopy regarding colorectal cancer (CRC) is poor. Adding a faecal immunochemical test (FIT) might improve triage of colonoscopy. The aim of this study is to assess the diagnostic utility of symptoms for the yield (CRC) of colonoscopy and to compare this with the diagnostic utility of FIT when offered to symptomatic patients. Methods: We performed a systematic review search for CRC as an outcome of colonoscopy in referred symptomatic patients and separately for CRC as an outcome in symptomatic patients with a positive FIT. Results: We included 35 studies, with almost 5 million symptomatic patients. In addition, we included nine prospective studies with a positive FIT in symptomatic patients, with more than 5000 patients. In a random effect model, the pooled sensitivity of colonoscopy in symptomatic patients was very low (25%). However, the pooled sensitivity in symptomatic patients with a positive FIT was 83% and the pooled specificity 77%. A total of 75 symptomatic patients (1.4%) had a false-negative FIT. Conclusion: Adding FIT in symptomatic patients seems useful for predicting CRC as an outcome of colonoscopy. FIT seems a potential tool for an improved triage of colonoscopy in symptomatic patients.

**Abstract:**

If Colorectal cancer (CRC) is detected and treated early, the survival rate is high. This is one of the reasons that population-based screening programs for the early detection of CRC using the faecal immunochemical test (FIT) started worldwide. These programs compete with regular colonoscopy programs and increase the waiting time for symptomatic patients. However, the literature has shown that the correlation between intestinal complaints and the gain of colonoscopy is poor. The aim of this study is to assess the diagnostic utility of symptoms for the yield (CRC) of colonoscopy and to compare this with the diagnostic utility of FIT when offered to symptomatic patients. Methods: We performed a systematic review search for CRC as an outcome of colonoscopy in referred symptomatic patients and separately for CRC as an outcome in symptomatic patients with a positive FIT. We searched systematically for clinical trials or observational studies in databases, followed by hand-searching of reference lists. We used random Meta-Disc to evaluate the diagnostic performance, using the exploration of heterogeneity with a variety of test statistics and by computing the pooled estimates. Results: We included 35 studies, with almost 5 million symptomatic patients. In addition, we included nine prospective studies with a positive FIT in symptomatic patients, with more than 5000 patients. Significant heterogeneity was found for every symptom and the outcome of colonoscopy in the effect size of sensitivity, specificity, positive likelihood ratio, negative likelihood ratio and diagnostic odds ratio. In a random effect model, the pooled sensitivity of colonoscopy in symptomatic patients was very low (25%). However, the pooled sensitivity in symptomatic patients with a positive FIT was 83% and the pooled specificity 77%. A total of 75 symptomatic patients (1.4%) had a false-negative FIT. Conclusion: Adding FIT in symptomatic patients seems useful for predicting CRC as an outcome of colonoscopy. FIT seems a potential tool for an improved triage of colonoscopy in symptomatic patients.

## 1. Introduction

Colorectal cancer (CRC) is the second most common cancer in the world [1,2]. Early detection and subsequent treatment of CRC increases the chance of survival. The 5-year survival rate in stage I is more than 90% and in stage IV just 10% [3,4]. Presumed predictive symptoms for CRC such as rectal bleeding, change in bowel habit and abdominal pain are non-specific and common in the general population [5,6,7,8,9]. The majority of these symptomatic patients (60–80%) do not have CRC. This makes it challenging to differentiate who to refer for colonoscopy. Subsequently, many patients are unnecessarily exposed to an unpleasant and invasive procedure with risk of complications [10,11,12]. Besides these risks, there are unnecessarily high costs involved [13].

The use of the guaiac faecal occult blood test (gFOBT) can reduce CRC mortality in asymptomatic population screening [14]. Additionally, major disadvantages have been identified (e.g., the test is not sensitive to small bleeds, specificity can be affected by diet or drugs, participant acceptance can be low, means of laboratory quality control are limited, and there is a fixed hemoglobin concentration cutoff determining positivity) [15]. For all these reasons, gFOBT seems obsolete to use for screening for CRC currently [16]. The current European guidelines recommended the new faecal immunochemical test (FIT), also called immunochemical fecal occult blood test (iFOBT), for CRC screening purposes [17]. Many studies have shown that FIT is superior to gFOBT for population-based CRC screening. The sensitivity for detecting CRC and advanced adenoma is higher; the participation rate is also higher. Another advantage of iFOBT is that the cut-off level of the hemoglobin concentration that defines a positive test can be defined [18]. The results of population screening with FIT in the Netherlands have shown a sensitivity of 65% and a specificity of 92% [19]. These programs compete with regular colonoscopy capacity and increase the waiting time for symptomatic patients.

If the expected yield of colonoscopy based on patient symptoms is poor and the sensitivity of FIT is high, the question arises whether FIT should be part of the diagnostic workup in symptomatic patients, to stratify patients on the waiting list based on expected yield of the colonoscopy.

The NICE guidelines recommend FIT for adoption in primary care to guide referral in people presenting with certain clinical signs and symptoms that may suggest colorectal cancer, but do not meet the following criteria:(1)aged 40 and over with unexplained weight loss and abdominal pain;(2)or aged 50 and over with unexplained rectal bleeding;(3)or aged 60 and over with iron-deficiency anaemia or changes in bowel habit; or tests show occult blood in their faeces.

In addition, CRC should be considered in adults with a rectal or an abdominal mass and in adults aged under 50 with rectal bleeding and any of the following unexplained symptoms: abdominal pain, change in bowel habit, weight loss, iron-deficiency anaemia [20].

According to these criteria, most symptomatic patients need to undergo a colonoscopy within two weeks after referral. It is questionable if this is really necessary.

The aim of this systematic review is to assess and discuss the diagnostic utility of symptoms for the yield (CRC) of colonoscopy and to compare this with the diagnostic utility of FIT when added to the workup of symptomatic patients.

## 2. Materials and Methods

We performed two literature searches. In the first literature search, studies were included that aimed to assess the yield of colonoscopy in terms of CRC when the indication for colonoscopy was based on the following symptoms: rectal bleeding, change in bowel habits, obstipation, diarrhea, abdominal pain, weight loss, iron-deficiency anemia or a palpable abdominal mass. In the second literature search, studies were included that aimed to assess the yield of colonoscopy in terms of CRC when the indication for colonoscopy was based on an positive iFOBT in addition to symptoms.

The systematic review followed the recommendations of the Preferred Reporting Items for Systematic Reviews and Meta-Analyses (PRISMA). The protocol has not been registered.

### 2.1. Search Strategy

Guidelines, PubMed, Medline, Tripdatabase and Cochrane databases were searched systematically from 1985 to May 2017 for clinical trials or observational studies, followed by hand-searching of reference lists. A combination of MeSH terms and text words was used. The full search strategy is given in additional file 1. Bibliographies and references of included studies, review articles and clinical guidelines were also searched. Only studies in English or Dutch were selected.

The patients in the included studies in both searches were referred for colonoscopy to exclude CRC. Studies that investigate the diagnostic accuracy of symptoms, signs and diagnostic tests in relation to CRC were used with a colonoscopy or barium enema as the reference standard.

All studies that used FIT in symptomatic patients were included. Papers that did not differentiate between CRC, polyps and/or IBD were excluded.

### 2.2. Quality Assessment

We extracted data of all papers regarding setting and design, study population, test characteristics and test results. Methodological quality was assessed with the quality assessment of a diagnostic accuracy studies (QUADAS) tool, which is recommended by the Cochrane Diagnostic Reviewers Handbook. This modified version consists of 14 items on methodological characteristics that have the potential to introduce bias. Items were scored as positive (no bias), negative (potential bias) or unclear. The QUADAS summary is shown in Table 1.

### 2.3. Data Extraction

The true positives, true negatives, false negatives and false positives of each individual symptom were extracted from the included articles. If these data were not mentioned, we tried to retrieve and calculate this with the necessary numbers. The study was excluded if we could not compute the data. Data extraction was conducted by one reviewer (BvD) and checked by a second reviewer (HJB).

### 2.4. Statistical Analysis

We used Meta-Disc to evaluate the diagnostic performance. The degree of heterogeneity of sensitivity specificity positive and negative likelihood and the diagnostic odds ratio among the studies was investigated using the likelihood ratio Chi-square test and the Q statistic. When the Q test was statistically significant, we changed from a fixed effect model to a random effects model. The I2 index was used for quantifying potential heterogeneity between the studies. In general, I2 = 25%, 50% and 75% corresponds with low, medium, and high heterogeneity, respectively. Stratified analyses were performed to investigate factors that could contribute to diagnostic performance across studies including prospective studies only, studies with only colonoscopy as reference standard, and studies including patient >30 years old.

## 3. Result

### 3.1. Publication Searching Results

We included 35 studies 26 prospective and 9 retrospective studies with a total of 4,833,056 symptomatic patients [21,22,23,24,25,26,27,28,29,30,31,32,33,34,35,36,37,38,39,40,41,42,43,44,45,46,47,48,49,50,53,54,55,56,58,59]. In addition, we included nine prospective studies with a positive FIT in symptomatic patients, with a total of 5296 patients [24,34,47,51,52,53,54,55,56]. The publication searching procedure is demonstrated in Figure 1 in a consort diagram.

Further study characteristics can be found in the Appendix B.

### 3.2. Statistical Heterogeneity

For every symptom, significant heterogeneity was found in the effect size of sensitivity, specificity, positive likelihood ratio, negative likelihood ratio and diagnostic odds ratio. These effect sizes were pooled by the random effect model.

### 3.3. Symptoms

Table 2 shows the pooled sensitivity, specificity, positive and negative likelihood and diagnostic odds ratio of the studied symptoms. The pooled sensitivity for CRC in symptomatic patients was 25%. A total of 14,159 symptomatic patients had been diagnosed with CRC, which means a true positive of 0.3%.

### 3.4. Symptomatic Patients and FIT

Table 3 shows the pooled sensitivity, specificity, positive and negative likelihood ratio for the FIT for patients with symptoms. Table 4 shows the Statistical heterogeneity evaluation. A total of 75 symptomatic patients (1.4%) had a false-negative FIT. Strikingly, only 35% of all patients with rectal bleeding also had a positive FIT.

## 4. Discussion

This study aimed to assess the diagnostic utility of symptoms for the yield (CRC) of colonoscopy and compared this with the diagnostic utility of FIT when added in symptomatic patients. Our study showed that using symptoms to predict CRC as an outcome of colonoscopy in symptomatic patients seems not useful. The pooled sensitivity for CRC in symptomatic patients is very low: 25%; this ranges from 5.5% in patients with an abdominal mass to 33% in patients with abdominal pain. On the other hand, the pooled specificity of a colonoscopy is 97%, ranging from 64% with diarrhea to 99% with weight loss. In our meta-analysis, only 0.3% of the symptomatic patients had been diagnosed with CRC.

On the contrary, the yield of colonoscopy if FIT is added in symptomatic patients is high. When the FIT is added in the diagnostic workup of a symptomatic patient, there is both an acceptable sensitivity (83%) and specificity (77%). FIT seems to be a potential tool for a better triage for colonoscopy in symptomatic patients.

The study is a review and meta-analysis of previous studies; the yield of colonoscopy when the indication is based on so-called “alarm features for CRC” is low.

The strength of this study is that we put it in a clear overview and the extensive meta-analysis indicates clear numbers. Another strength and novelty of this review is that we did a meta analysis on the yield of colonoscopy when a positive FIT is added to symptoms.

The study methods followed the traditional scheme for systematic reviews. A broad selection of symptoms was chosen to ensure all relevant studies were included. We did not use many exclusion criteria, so we included a wide variety of studies. The strengths of this research also entail some implications. Because of the wide variety of studies, the total study period is very long. We chose to focus on single symptoms, because otherwise too many combinations are possible, leaving fewer studies to compare. Although it seems plausible that patients with more symptoms have a higher chance of CRC, other symptoms may actually have been present but were not reported in the included studies.

Most of the symptoms are very subjective and no clear symptom description was given. For the sake of clarity in this paper, and to include as many studies as possible per symptom, we did not distinguish between different terminologies of the same symptom. Because we included a wide variety of studies, there was considerable heterogeneity between studies. The larger studies are all from electronic databases; their results are not directly comparable to smaller studies. The heterogeneity may also be due to the unclear symptom description and differing severity and variations in referral rates. Additionally, and probably most importantly, every included study showed low sensitivity for all the symptoms. The low sensitivity is probably explained by the fact that the symptoms studied are common in the general population and may have many other causes besides a CRC (3—7). Doctors should be aware that the yield of colonoscopy in terms of CRC when the indication is only based on symptoms is low. The majority of these patients are unnecessarily exposed to an unpleasant and invasive procedure with a risk of complications and needless costs involved [13]. Furthermore, waiting lists for colonoscopy are becoming longer since the introduction of the population screening for CRC [13].

The NICE guideline was adjusted in 2017, which recommends FIT for people with unexplained symptoms that do not meet the criteria for a suspected cancer pathway referral. The recommendation after a positive FIT is to perform a colonoscopy within two weeks [20]. By these criteria, most symptomatic patients are covered by the suspected CRC pathway. Is it really necessary to consider all of these patients as suspected for CRC? 

## 5. Conclusions

We describe a true-positive rate of CRC as an outcome of colonoscopy in 0.3% of these patients. We would suggest using the FIT as a triage tool for colonoscopy, to increase the sensitivity of symptoms. Patients with symptoms and a negative FIT are probably better off with a consultation of a gastroenterologist before planning for colonoscopy, as there may well be another underlying cause. Further research is needed to test this triage system.

## Figures and Tables

**Figure 1 cancers-15-02011-f001:**
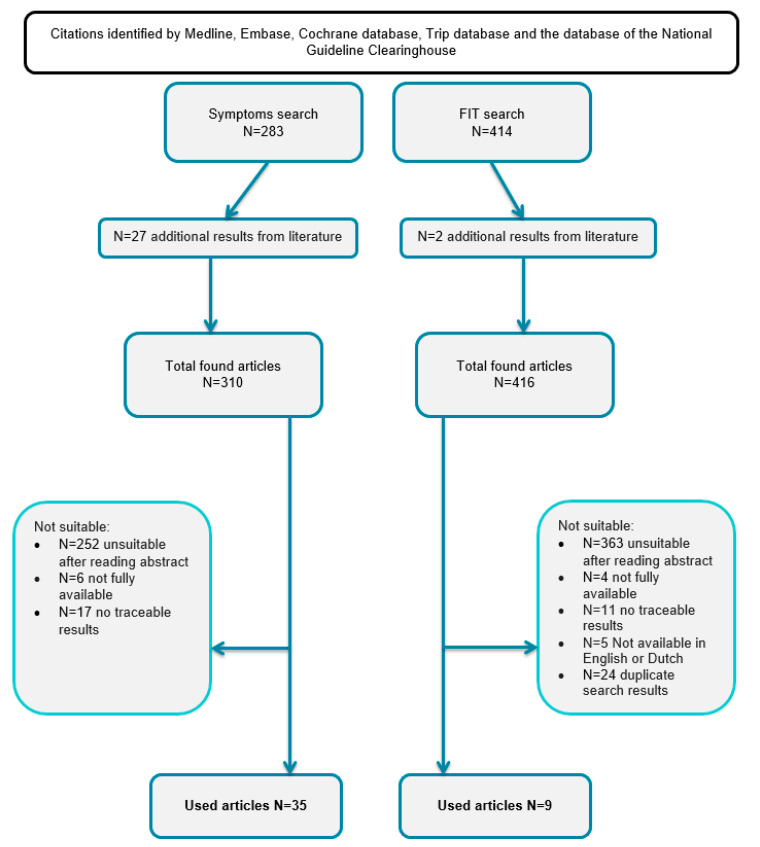
The publication searching procedure.

**Table 1 cancers-15-02011-t001:** The QUADAS method.

	1	2	3	4	5	6	7	8	9	10	11	12	13	14
Bafandeh 2008 [21]	+	+/−	+	+	+	+	?	+	+	−	−	+	+	+
Bjerregaard 2006 [22]	+	+	+	+	+	+	+	+	+	+/−	+/−	+	?	+
Brewster 1994 [23]	+	+/−	−	+	+	−	+	−	+	+	+/−	+	−	−
Farrands 1985 [24]	+	+/−	+/−	+	+	+/−	+	+/−	+	+	+/−	+	?	−
Selvachandran 2002 [25]	+	+/−	+	+	+	+/−	+	+	+	+	+/−	+	?	+
Steine 1994 [26]	NB													
Tan 2002 [27]	+	+/−	+	+	+	+	+	+/−	+	+/−	+/−	+	?	−
Tate 1988 [28]	+	+/−	+	+	+	+/−	+	+/−	+	−	+/−	+	?	+
Thompson 2007 [29]	+	+	+/−	+	+	+	+	+	+	+	−	+	?	−
Thompson 2008 [30]	+	+	+/−	+	+	+/−	+	+/−	+	+	+/−	+	?	−
Zarchy 1991 [31]	NB													
Panzuto 2003 [32]	+	+	+	+	+	+/−	+	+	+	+	+/−	+	?	+
Curless 1994 [33]	+	+	+	+	+	−	+	+/−	+	−	+	−	?	+
Jensen 1993 [34]	+	+/−	+/−	+	+	+/−	+	+/−	+	+/−	−	+	?	?
Patel 2016 [35]	+	+/−	+	+	+	+/−	+	+/−	+	−	+/−	+	?	?
Cheong 2000 [36]	+	+/−	+	+	+	+/−	+	+/−	+	+/−	+/−	+	?	?
Hipsley-Cox 2012 [37]	+	+	+	?	+	+/−	+	+	+/−	−	?	+	−	+
Simpkins 2017 [38]	+	+	+	+	+	+/−	+	+	+	+	+	+	?	+
Hamilton 2009 [39]	+	+	+	+	+	+	+	+	+	+	+/−	+	?	?
Koning 2015 [40]	+	+	+	+	+	+/−	+	+	+	+	−	+	?	?
Hamilton 2005 [41]	+	+	?	?	?	?+	+	+/−	?	−	?	+	−	+
De Bosset 2002 [42]	+	+	+	+	+	+	+	+	+	+	−	+	?	?
Cai 2015 [43]	+/−	+/−	+	+	+	+/−	+	+/−	+	+	−	+	?	?
Pepin 2002 [44]	+	+	+	+	+	+	+	+/−	+	+/−	+/−	+	?	?
Flashman 2004 [45]	+/−	+	+	+	+	+/−	+	+	+	+	+	+	?	+
Du toit 2006 [46]	+	−	+	+	+	−	+	+/−	+	+	−	+	?	?
Nakama 2000 [47]	+	+	+	+	+	+	+	+	+	+	−	−	?	?
Wauters 2000 [48]	+	+/−	+	+	+/−	+/−	+	+/−	+/−	+/−	+/−	?	?	−
Ahmet 2005 [49]	+/−	+	+	+	+	−	+	+	+	+/−	+/−	+	−	+
Brenna 1990 [50]	+	+/−	+	+	+	−/+	+	−	+	+	−	+	−	+
Mc donald 2013 [51]	+	+	+	+	+	+	+	+	+	?	?	+	?	−
Elias 2016 [52]	+	+	+	+	+	+/−	+	+	+	+	+	+	?	+
Hogberg 2017 [53]	+	+	+	+	−	−	+	+	+	+	−	+	?	+
Mowat 2016 [54]	+	+/−	+	+	+	+	+	+	+	+	+	+	?	+
Cubiela 2014 [55]	+	+	+	+	+	+	+	+/−	+	+	+	+	?	+
Cubiela 2016 [56]	+	+	+	+	+	+	+	+	+	+	+	+	?	+
Rodriguez 2015 [57]	+	+	+	+	+	+	+	+	+	+	+	+	?	+
Law 2014 [58]	+	+/−	+	+	+	+	+	+/−	+	+	?	+	?	+

+ = no bias; − = bias −/+ = potential bias; ? = bias unclear. 1 = valid selection, representative patients, 2 = selection clearly described 3 = adequate reference test 4 = target condition did not change between tests, 5 = all/random selection received verification via reference test, 6 = all received same test, 7 = index test not part of reference 8 = index test described in detail 9 = reference test described in detail. 10 = blinded to reference standard, 11 = blinded to index test, 12 = clinical data available as normal, 13 = no missing/uninterpretable data 14 = withdrawals explained.

**Table 2 cancers-15-02011-t002:** The pooled sensitivity, specificity, positive and negative likelihood and diagnostic odds ratio of the studied symptoms.

	Pooled Sensitivity	Pooled Specificity	Pooled Likelihood +	Pooled Likelihood −	Pooled DOR
Changed bowel habit	0.235 (0.226–0.244)	0.974 (0.974–0.973)	1.603 (1.194–2.151)	0.841 (0.773–0.914)	1.979 (1.158–3.382)
Diarrhea	0.192 (0.182–0.202)	0.635 (0.625–0.644)	0.747 (0.278–2.008)	1.119 (0.684–1.834)	0.650 (0.139–3.050)
Obstipation	0.266 (0.254–0.277)	0.888 (0.885–0.891)	1.168 (0.754–1.809)	1.022 (0.900–1.161)	1.177 (0.698–1.986)
Anemia	0.285 (0.274–0.297)	0.985 (0.985–0.984)	2.661 (1.911–3.704)	0.818 (0.707–0.947)	3.490 (2.523–4.826)
Abdominal pain	0.329 (0.319–0.340)	0.741 (0.743–0.738)	1.176 (0.825- 1.676)	1.006 (0.894–1.133)	1.161 (0.703–1.918)
Weight loss	0.116 (0.110–0.123)	0.986 (0.987–0.986)	2.358 (1.684–3.300)	0.902 (0.863–0.943)	2.741 (1.835–4.094)
Rectal blood loss	0.313 (0.305–0.322)	0.963 (0.963–0.963)	2.037 (1.286–3.227)	0.837 (0.767–0.913)	2.501 (1.337–4.677)
Abdominal mass	0.055 (0.030–0.090)	0.969 (0.963–0.974)	1.780 (0.798–3.969)	0.991 (0.950–1.033)	1.018 (0.364–2.843)
Cumulative symptoms	0.248 (0.244–0.251)	0.972 (0.971–0.972)	1.620 (1.356–1.936)	0.923 (0.896–0.951)	1.792 (1.389–2.311)

**Table 3 cancers-15-02011-t003:** The pooled sensitivity, specificity, positive and negative likelihood ratio for the FIT for patients with symptoms.

	Pooled Sensitivity	Pooled Specificity	Pooled Likelihood +	Pooled Likelihood −	Pooled DOR
FIT	0.830 (0.792–0.863)	0.765 (0.755–0.775)	3.886 (2.640–5.721)	0.155 (0.086–0.278)	27,025 (18,509–39,459)

**Table 4 cancers-15-02011-t004:** Statistical heterogeneity evaluation FIT.

Effect Size	Chi Square	I2%	*p*
Sensitivity	63.19	82.6	0.000
Specificity	725.01	98.5	0.000
Positive likelihood ratio	589.80	98.1	0.000
Negative likelihood ratio	39.82	72.4	0.000
Diagnostic odds ratio	12.80	14	0.307

## Data Availability

There are no data created.

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
