# Peer review of "Possible Value of Faecal Immunochemical Test (FIT) When Added in Symptomatic Patients Referred for Colonoscopy: A Systematic Review"

_cancers, 2023, doi:10.3390/cancers15072011_

Round 1

Reviewer 1 Report

thank you for the opportunity to review this literature review. It is usually recommended to perform the immunological screening test in asymptomatic patients, without family history and aged over 50 years, in order to diagnose colorectal cancer at an early stage. the authors propose to perform this immunological test in case of symptoms in order to better target the indications of total colonoscopy and to increase its reliability. 

The results of this meta-analysis need to be qualified because of the heterogeneity of the studies, the very long inclusion period (>30 years) and the lack of details regarding symptomatology. It would be interesting to look at the results of subgroup analyses by design (prospective vs retrospective studies) and to assess the impact of study period on the results. Were the authors able to analyse the reliability of colonoscopy and/or immunoassay according to the category of symptoms presented by the patients? 

Author Response

Thank you very much for the review of our manuscript  and for your clear comments.

We agree with your comment with respect to heterogeneity of the studies and long inclusion period.  We chose to include a wide variety of studies for a long study period, to be complete as possible. In the discussion part of the study we have mentioned this point extensively. However as all included studies showed low sensitivity for all the symptoms. We are the opinion that this adds to the strength of the study.

It could be of interest for better triage to do subgroup analyses in patients with certain symptoms to look at the yield of the FIT test. However this would not change the conclusion of our manuscript.

Reviewer 2 Report

This is an exciting paper about a topic that would interest the journal's readers. The study addresses the potential clinical value of FIT in symptomatic patients referred for colonoscopy, and it is pretty well-designed, and the methods are adequately used. Although there is no major novelty to the field with this study and several significant limitations (already highlighted by the authors), it can be considered for publication because it may add value to the current literature.

Only a few minor issues:

Please comment on the results that only 35% of all patients with rectal bleeding have had a positive FIT.

In line 50, please consider changing aspecific to non-specific.

In line 51, please replace differentiatie with differentiate.

Author Response

Thank you very much for reviewing our manuscript and for your clear comments

Thank you for your compliments. We have processed the minor issued you mentioned in our article.

Furthermore about your question of the 35% positive FIT with rectal bleeding, I want to refer to this article: Hicks G, D'Souza N, Georgiou Delisle T, Chen M, Benton SC, Abulafi M; NICE FIT steering group. Using the faecal immunochemical test in patients with rectal bleeding: evidence from the NICE FIT study. Colorectal Dis. 2021 Jul;23(7):1630-1638. doi: 10.1111/codi.15593. Epub 2021 Mar 15. PMID: 33605522.

This is a citation from this study:

This study showed that f-Hb is not always detected in patients with RB. Detectable f-Hb (f-Hb >2 µg/g) was present in only 44.1% of patients with RB, compared with 33.9% of NRB patients (p < 0.05). While significant, this may be attributed to the higher prevalence of SBD in the RB group. Undetectable f-Hb in patients with RB can be explained by sporadic RB in both significant and nonsignificant bowel disease. Moreover, bleeding from an anorectal source will most likely coat the outside of the stool sample and may not be distributed throughout the bulk of the stool from where a sample is obtained.

Round 2

Reviewer 1 Report

 Accept in present form